# Mothers as facilitators for a parent group intervention for children with Congenital Zika Syndrome: Qualitative findings from a feasibility study in Brazil

**Tracey Smythe**[1☯]*, **Monica Matos**[2☯], **Julia Reis**[3☯], **Antony Duttine**[1‡], **Silvia Ferrite**[4‡], **Hannah Kuper**[1‡]

**1** International Centre for Evidence in Disability, London School of Hygiene & Tropical Medicine, Keppel Street, London, United Kingdom, **2** Collective Health Institute, Universidade Federal da Bahia, Salvador, Brazil, **3** Department of Child Psychology, Rehabilitation Institute of Bahia, Salvador, Brazil, **4** Department of Speech and Hearing Sciences, Federal University of Bahia, Salvador, Brazil

☯ These authors contributed equally to this work.
‡ These authors contributed equally to this work
* tracey.smythe@lshtm.ac.uk

**Data Availability Statement:** Based on restrictions imposed by the London School of Hygiene & Tropical Medicine (LSHTM) Ethics committee, data

## Abstract

### Background

The Zika virus outbreak in Brazil (2015–2016) affected thousands of children who were born with Congenital Zika Syndrome (CZS). Families play an important role in their care of children with complex needs, yet their knowledge, experience and skills are rarely harnessed in existing interventions to best support these families.

### Objective

This study explores the use of mothers as facilitators for a community-based group intervention for children with CZS and their caregivers in Brazil.

### Methods

Four facilitators were trained to deliver the 10-week intervention called "Juntos". Two were mothers of a child with CZS ("expert mothers") and two were therapists (speech therapist and physiotherapist). The intervention was delivered to three groups, generally including 8–10 caregivers. Two researchers, who were psychologists, observed the groups and held focus group discussions at the end of each session. They undertook semi-structured interviews post intervention with a purposive sample of caregivers, and with the facilitators. Observation notes were collated and summarised. Transcripts were transcribed and thematically analysed using five elements to assess feasibility: acceptability, demand, implementation, practicality and adaptation.

associated with this study will not be made freely available, as the small number of children with CZS makes data potentially identifying. The DOI for our data files titled 'The feasibility of a group intervention for children with congenital zika syndrome – qualitative dataset' is https://doi.org/10.17037/DATA.00001762. Applications for access to the raw data for this study should be made via this DOI, outlining the purpose of the proposed analyses and the data requested. These applications will be reviewed by the LSHTM's data access committee, and if accepted, the requested data will be shared.

**Funding:** The development and testing of the intervention for children with Congenital Zika Syndrome were supported by a grant from Wellcome Trust and the Department for International Development (https://wellcome.ac.uk/) to HK: grant code 206719/Z/17/Z. The funders had no role in the design of the study; in the collection, analyses, or interpretation of data; in the writing of the manuscript, or in the decision to publish the results.

**Competing interests:** I have read the journal's policy and the authors of this manuscript have the following competing interests: One of the authors (AD) joined the Pan American Health Organisation (PAHO) during the research period. Work on the research study was undertaken outside and separate to his PAHO duties. This does not alter our adherence to PLOS ONE policies on sharing data and materials.

## Results

The use of expert mothers as facilitators was considered to be acceptable and there was demand for their role. Their experiential knowledge was viewed as important for sharing and learning, and supporting and encouraging the group. The intervention was delivered with fidelity by the expert mothers. The practicality of the intervention was facilitated by holding the group sessions in the community, providing transport costs to facilitators and participants, paying expert mothers and therapist facilitators equally and supporting the expert mothers through a mentorship programme. Equal payment with the therapist enabled the expert mothers to better facilitate the groups, through increased confidence in the value of their role. Adaptation of the intervention included development of video resources and mentoring guidelines.

## Conclusion

The use of expert mothers as facilitators of caregiver groups provides a unique approach to harness the knowledge, experience, and skills of families to provide care, and is likely to be feasible in similar contexts.

## Introduction

Developmental disabilities affect at least 50 million children under the age of 5 years globally, and are a major contributor to child and adult morbidity in low and middle-income countries (LMICs) [1]. Children with developmental disabilities have multiple impairments (e.g. cognitive, physical, visual) that have a long-term influence on their health and development [2]. Families play an important role in care for these children, and caregivers of children with disabilities often experience high levels of stress, anxiety, depression, physical exhaustion, and discrimination [3–8]. These experiences contribute to decreased quality of life compared to caregivers of non-disabled children, and may result in reduced effectiveness of parenting [9, 10]. Evidence is lacking on how best to support these families, particularly those living in resource limited settings where healthcare providers with the appropriate expertise may be lacking. Support programmes in partnership with health professionals are increasingly being used to try to fill these gaps, but they often have a top-down approach, rather than drawing on the knowledge, experience and skills of family members.

The World Health Organisation (WHO) recommends a range of interventions for children with or at risk of developmental disabilities. These generally include a focus on empowerment of caregivers and a shift from child-centred to family-centred care, to provide optimal stimulation for development in a safe, stable and nurturing environment [11]. Family-centred care is an approach in which families are recognised as the experts on their child, and work with service providers to make informed decisions about their child's care. There is a growing body of evidence on how best to offer family-centred care in lower resourced settings, including through participatory peer learning [12]. Peer support interventions are hypothesised to work by increasing the amount of social support available to parents and caregivers, and providing that support in a form which is most useful and acceptable to participants [13]. As an example, women's groups that practice participatory learning and action have been shown to improve maternal and child health and empower women [14], and are recommended by the WHO to reduce newborn mortality and improve health in low resource settings [16]. With respect to

caregivers of children with developmental disabilities, participatory peer learning in groups provides caregivers opportunities to practice activities with their child and receive feedback, and has greater benefits compared with providing parenting information only [11]. Support groups provide positive benefits to child development and wellbeing, and family functioning [14], improve caregiver understanding, confidence and self-esteem, and reduce self-blame [15]. Additionally, support groups may offer an important social safety net for caregivers who are excluded in their communities [15].

One context in which peer support group interventions could be relevant is for the care of children affected by the Zika Virus (ZIKV). Since late 2015, there have been more than three thousand cases of microcephaly suspected of being related to ZIKV in Brazil [16–18]. Congenital infection with ZIKV is linked to other abnormalities besides microcephaly, including neurological conditions, ophthalmic abnormalities, hearing loss and bone and joint disorders [19–21], now collectively called Congenital Zika Syndrome (CZS). Children with CZS are likely to have complex intellectual, physical and sensory impairments over their lifetime [22]. Additionally, a wider spectrum of developmental impairments may yet manifest in children who were exposed to ZIKV in utero, but do not have CZS, as they continue their development [23]. Provision of family and supportive services is essential to meet the broader needs of these children and caregivers, and to complement clinical and other services available in Brazil.

We therefore developed a group intervention ("Juntos", meaning 'together' in Portuguese) to provide psychosocial support and improve the skills of caregivers of children with CZS in Brazil, to optimally care for their child [24, 25]. The Juntos programme consists of ten sessions offered over a period of 3 months held in the local community. Each session includes icebreaker activities, practical sessions and group discussions, and a psychological support component, and lasts approximately 4 hours. The content of the programme takes learning from health concerns that affect children with cerebral palsy (CP) [26], which is a similar developmental disability to CZS. The programme includes participatory learning about care practices, such as feeding, positioning, communication, everyday activities, play and early stimulation, in addition to disability rights and inclusion (Table 1). Every session includes an emotional support activity to provide a safe environment in which to stimulate open and supportive discussion between caregivers about their successes and difficulties. Programme material is available from www.ubuntu-hub.org

The existing caregiver support programmes for children with CP relied on therapists as facilitators, but within Juntos we pilot-tested the use of a parent of child with CZS ("expert mother") working alongside a therapist facilitator. This approach was used to encourage a more participatory process and atmosphere of sharing between the caregivers. There were concerns about the acceptability and practicality of this approach, and we therefore aimed to explore the feasibility of the use of mothers as facilitators for the community-based group intervention for children with CZS and their caregivers in Greater Salvador, Bahia.

## Materials and methods

Ethical approval was gained from Instituto de Saúde Coletiva—ISC/UFBA Ethics Ref 2.369.348, Instituto Fernandes Figueira—IFF/ FIOCRUZ—RJ/MS Ethics Ref 2.183.547 and LSHTM Ethics Ref 13608. Written informed consent was acquired from all participants.

### Study design

This study is part of a larger pre post intervention design in Greater Salvador and Rio de Janeiro, which has been described in detail previously [24]. This qualitative study was undertaken between August 2017 and May 2018 in three municipalities of Greater Salvador, Bahia.

**Table 1. Juntos intervention module topics.**

| Module number and title | Topics covered |
| --- | --- |
| 1: Introduction | About the programme |
| | Information about Zika and Congenital Zika syndrome |
| | How to find information |
| | Personal stories |
| 2: Our child | Introducing your close family and friends |
| | Development milestones for young children |
| | Determining your child's progress |
| | Managing irritability and crying |
| 3: Positioning and moving | How to position children who need assistance |
| | How to assist children to learn to move |
| 4: Eating and drinking | Feeding challenges |
| | Practical skills to address challenges for your child |
| 5: Communication | Importance of communication |
| | Practical advice to help your child communicate |
| 6: Play and early stimulation | Importance of play for children to develop and learn |
| | Early stimulation |
| | Making simple toys |
| | Inclusion of play in the family and broader community |
| 7. Everyday activities | How to use everyday activities to help your child develop |
| | Managing seizures |
| 8. Uniting our voices | Understand the context of disability rights |
| | Education |
| | Communicating with your health team |
| | Advocating |
| 9. Our community | Who is in your community |
| | Common barriers to inclusion |
| | Addressing negative attitudes and exclusion |
| | Social Activity |
| 10. Next steps | Summing up |
| | Planning next steps for yourself and the group |

Data were collected from observation of the caregiver groups (n = 30), focus group discussions with caregivers (n = 30), and semi-structured interviews with group facilitators (n = 4) and purposively selected caregivers (n = 9).

## Participants and setting

Participants were caregivers of children with neurologist-confirmed CZS in Great Salvador, Bahia. For the purposes of this study, we used CZS to describe any child with impairments that can be directly attributed to Zika. The impairments in children ranged from mild cognitive, communication or functional skill delay to severe delay in all three developmental categories. The average group size was 8–10 caregivers. Each group met weekly for approximately four hours for ten sessions.

The groups were facilitated by one expert mother paired with one therapist (speech therapist or physiotherapist); in total two mothers and two therapists were included. The facilitators were identified and selected by the study site co-ordinator. Selection characteristics included (i) a similar socioeconomic level to participants, (ii) having a child with a pattern of severity of

CZS, (iii) a willingness to help other families, (iv) an active participant in disability rights, and (v) tolerance to personal differences and perspectives (e.g. religion). Both expert mothers were purposively selected by the site co-ordinator, as they had a similar socio-economic level as the group participants, were literate and had completed high school at a minimum as they were required to follow a facilitator manual to deliver the intervention. The hypothesised role of the expert mother was to facilitate a participatory and equal atmosphere, and encourage the sharing of learning between caregivers. For example, the expert mother led the emotional support activity to help to stimulate open and supportive discussion between caregivers about their successes and difficulties in the previous week. The Juntos materials were also developed to include videos of expert mothers and fathers demonstrating techniques such as brushing teeth and play. The role of the therapist was to facilitate the technical aspects of skill acquisition and practice of techniques, such as feeding positions. The therapists and expert mothers completed a joint standardised 5-day facilitators training programme to prepare them for the delivery of Juntos. The expert mother was paid the same as the facilitator therapist (2,100BRL/month, approximately USD500). Transport costs were reimbursed all facilitators and all participants.

## Data collection

The reasons for undertaking the research were explained, and data were collected by two female Brazil based research assistants (psychologists). Neither of the psychologists worked in the clinical area or had prior knowledge of the participants. Three techniques were used for the collection of qualitative data. First, participant observations of all ten sessions for each of the three groups were made during each session by the research assistants. An observation checklist was used to assess fidelity of delivery of the intervention by the facilitators. In addition, observation of the ice-breaker activities, practical sessions and group discussions provided an opportunity to examine the interaction of the expert mothers with the participants and facilitator therapist during each session. The psychologists observing the sessions wrote detailed field-notes, and an Excel file was completed for each session. Second, the research assistant (one per site) facilitated focus group discussions related to the content and processes of the session (approximately forty-five minutes) at the end of each session. The research assistant took comprehensive notes of the discussion with the participants. Third, semi-structured interviews were undertaken by the research assistants with both facilitators of the group and a purposively selected sample of participants, to allow for triangulation of data.

Three participants per group were selected after the completion of the ten intervention sessions. Participants were selected at the discretion of the researchers to reflect a broad range of perspectives (e.g. caregivers of children with different severities of disabilities, different caregivers that included mothers, fathers and grandmothers, caregivers of different ages and reflecting a geographic spread). The interview guide was piloted for understanding (S1 and S2 Tables) and interviews lasted on average 1 hour, ranging between 50 minutes and 1 hour 20 minutes. Interviews were designed to explore participants' motivation for attending the intervention, their views around how they were approached to participate in 'Juntos', how they viewed the programme itself, and aspects around engagement were explored. The interviews were audio recorded by digital sound recorder. Focus groups and interviews were conducted in Portuguese.

## Data management and analysis

Observational data and comments that related to the role of facilitators from focus group discussions were collated in an Excel sheet (Microsoft Excel 2000 (Microsoft Inc., Redmond, Washington) after each group intervention by one lead UK based researcher, a specialized

paediatric physiotherapist trained in qualitative and quantitative techniques (TS). The two research assistants (MM and JR) transcribed and translated the audio-recorded data from the semi-structured interviews.

We assessed the feasibility of the participatory group intervention using a framework based on a model proposed by Bowen et al (2010) [27]. Specifically, we focussed on the facets of acceptability, demand, implementation, practicality and adaptation of the delivery of the programme by expert mothers, in collaboration with a therapist. Acceptability considers how the recipients react to the inclusion of expert mothers. Demand for the intervention is indicated by the documenting of activities by the expert mother in the caregiver groups. Implementation concerns the extent, likelihood, and manner in which facilitation by the expert mother can be fully implemented as planned and proposed, often in an uncontrolled design. Practicality considers the extent to which facilitation by the expert mother can be implemented when resources, time, commitment, or some combination thereof are constrained in some way. Adaptation focusses on how the role of the expert mother may need to be adapted for a new situation. The remaining three areas outlined in the Bowen model of limited efficacy, expansion and integration are to be explored in future studies.

Words and phrases relating to the effect of having an expert mother facilitate the intervention were identified and coded to the feasibility framework. They were discussed between the two research assistants for agreement and checked and verified by TS. Consensus on coded phrases was gained through discussion. We undertook a narrative synthesis of the findings. Illustrative quotes are presented in the findings. We reported the results according to the consolidated criteria for reporting qualitative research (COREQ) [28], which is a 32-item checklist.

## Results

Three Juntos groups were convened in Greater Salvador (in Simões Filho, Lauro de Freitas, Camaçari) between August 2017 and May 2018. A total of 25 families and their children enrolled across the three groups. Thirty-eight participants (mother, father or other caregiver) attended at least one session. The two research assistants undertook 120 hours of observation.

### Acceptability

Support from expert mothers was considered to be acceptable and highly valued by the participants in this study. Both participants and facilitators talked about the benefits of including the expert mother and the role of the expert mother was viewed as acceptable in two ways; (1) promoting sharing and learning from each other, and (2) supporting and encouraging each other.

**(1) Sharing and learning from each other.** The research assistants observed that inclusion of the expert mothers contributed to an environment in which participants shared their own experiences and listened to others facing the same challenges. Expert mothers' knowledge and expertise from first-hand experiences was shared with others in similar situations, and an example of sharing from the perspective of the expert mother includes:

> "*I experienced something really incredible when a mother did not believe that her daughter was capable of doing something and, at that moment, I remembered that my son had had the same difficulty. I knew that her child was capable, so I shared strategies that I learnt for my child and she was able to do it.*" (Expert mother 01)

Expert mothers led by example to increase the inclusion of other family members and their communities to assist with care of their child. One expert mother required support from her husband to facilitate a group and she explained to the group:

"*to be here [facilitating the group intervention] I had to train my husband*" (Expert mother 02)

This example demonstrated to other caregivers what may be possible when seeking support from other family and community members.

Caregivers and expert mothers reported finding a shared social identify. This was evident through the shared experience of their children's unpredictable medical needs, which was faced by the expert mothers and the participants. For example, an expert mother was unable to attend one session due to her child being ill:

"*I cannot be here today because I have to be in hospital with my child*" (Expert mother 01)

Fostering a shared sense of identity led caregivers to meet outside of the support programme after the three months concluded, expanding their social and support network.

**(2) Supporting and encouraging each other.** The expert mothers were regularly observed to encourage parents that "you know better than you believe". Through the process of participatory peer learning, one expert mother told us:

"*By learning and acting as an expert mother, Juntos enabled me to find out about so many new things and people, who will forever remain part of my story. We had so many questions, and together we mothers were heard and we listened to each other. It was a very important change because normally families believe that they should just listen to professional therapists.*" (Expert mother 01)

The expert mothers appreciated how difficult it could be to meet the needs of a child with a disability. Participants reported that acceptance from the expert mothers helped to increase their self-belief and confidence in their own ability to care for their child. This was facilitated through being seen as on an 'equal level':

"*At the first sessions I felt like a mere spectator. I was only there to "learn" to handle my daughter. But, what does that mean, "learn"? Does it mean that I, as her carer, who is with her every second of the day, has nothing to offer? . . .we ran the sessions so that the families had equally important roles.*" (Expert mother 02)

The benefits of including the expert mother and reducing the traditional medical hierarchy were also articulated by the therapist facilitator. One facilitator therapist reported change in how she provided support to families:

"*Being a facilitator is not easy, you need to be available to care. . .sharing the facilitator space with a mother enabled me to get closer to a reality that I had only experienced from a distance. Hearing from other carers and sharing so much knowledge, which only they have, was transforming.*" (Facilitator therapist 02)

The expert mothers were viewed as acceptable to participants as they encouraged the provision of mutual support. This provided a strong motivation for participants to offer the same support to other caregivers, which they had benefitted from themselves:

"*No one has asked before about my hopes or dreams. Today I see if I am well then I can take better care of my child. Others should know this too. We must take care of ourselves. . .it is important to help us understand this. I now tell others about this often.*" (Participant 04)

There was no reported objection from other mothers about payment of expert mothers as facilitators. Limited acceptability may occur when caregivers are concerned with the comparison between their own and the child of the expert mother. For example, if participants consider the child of expert mothers much less affected by CZS compared with their own child. However, this was not experienced within these groups.

## Demand

There was a demand for expert mothers expressed by the other caregivers, and participants reported valuing the opportunity to share personal experiences and problem solve together with someone who understood their circumstances. The expert mothers provided an example of what may be possible through their lived experience, which indicates the way the intervention was delivered was appropriate and accessible to caregivers. This experience was underpinned by the importance of building hope and confidence. One participant acknowledged:

> "*if she is telling me, then I can do it.*" (Participant 07)

Additionally, as the facilitator pairs worked together over time, the value of the expert mothers became more clear to the therapist facilitator. The therapist facilitators voiced wanting to continue facilitating groups of parents alongside an expert parent:

> "*Working together with a mother facilitator, my senses are much more attentive to each family's story, and these stories have acted on me and changed me, both personally and professionally.*" (Facilitator Therapist 02)

Expert mothers were viewed to improve the understanding of the need of families and the professional abilities of the therapist facilitator to address these needs.

## Implementation

The intervention content and processes were delivered with fidelity by the expert mothers; the content (e.g. topics covered from module one to ten, the emotional support activity) and processes (e.g. participatory learning, adult learning techniques) of the intervention were used to a large extent by the expert mothers. The expert mothers enacted their expected role within the group. The research assistants observed that they facilitated a participatory and equal atmosphere, and encouraged the sharing of learning between caregivers. The expert mothers filled their planned role (described above) and this differed from the role of the therapist, who facilitated the technical aspects of skill acquisition, such as feeding positions. However, expert mothers described a tension in their role as a facilitator:

> "*There are some moments where I freeze. It is more difficult that I thought it would be, to be a mother and a facilitator, because I keep pushing myself to act like a therapist during the theoretical moments.*" (Expert mother 01)

Both facilitators contributed to practice of techniques. The relationship between the facilitator therapist and the expert mother developed over time through the delivery of the ten session programme and this relationship was sustained by enthusiastic therapists who advocated for family centred services.

Interviewer: "*What would you like to do differently in the next session?*"

*"We would like to have a meeting, before the session, to prepare our work together. This will help us to feel more confident to lead the next session."* (Facilitator therapist 01)

However, changes to the on-going support through increased access to mentors was required. For example, the expert mothers led the emotional support activity as part of every session, but they did not have a mental health training. The observing research assistants, who were trained psychologists, identified the need to provide regular psychological support to the expert mothers after delivering each session. The expert mothers were therefore provided with weekly mentoring by the psychologists. This input deviated from the limited support that was initially planned.

## Practicality

The practicality of the intervention was facilitated by holding the group sessions in the community, providing transport costs to the expert mothers, paying expert mothers and therapist facilitators equally and supporting the mothers through a mentorship programme. However, certain challenges were identified in the practicality of including mothers as facilitators. There were two occasions when an expert mother was unable to attend. The weekly sessions were held on different week days and the other expert mother was able to assist with one session. In the other session, it was run only by the facilitator therapist. The expert mother in these cases was unable to attend due to her child being ill. Over the three-month period, the majority of caregivers missed at least one session due to illness of their child. Fifteen (60%) families attended seven or more modules. Four families participated in all sessions. Module nine, titled 'Our Community', which includes a community day and celebration, was the best attended (group one n = 9; group two n = 7; group three n = 6). The 'Play and Early Stimulation' module delivered in week six was the least attended (group one n = 4, group two n = 4, group three n = 3).

The group sessions were designed to be held over two to three hours, and typically took approximately four hours. Observation notes suggest that the logistics of delivering the intervention could be challenging, with sessions often starting behind schedule and all the content was not covered in some sessions. As the group facilitators became more familiar with the content, the delivery of the intervention did not exceed the allocated time.

With regards finance, it appeared that equal payment enabled the expert mothers to better facilitate the groups, being more confident of the value of their role in a pair with the therapist. However, the decision to pay expert mothers and therapist facilitators the same amount was initially viewed as unfavourable by the administration team due to precise costing guidelines provided for therapists and relevant experience, with the suggested payment for expert mothers as 1,500BRL/month, approximately USD360. The estimate of salary cost for one therapist to deliver 10 group sessions over three months is 6,300BRL (approximately USD1,500) and the incremental cost of including an expert mother would be 6,300BRL (approximately USD1,500) per programme.

The success of the facilitator partnership is likely dependent on both personality and perspective of the pair. Selection characteristics of expert mothers included a similar socioeconomic level to participants and having a child with a severe pattern CZS, with personal characteristics of consideration and understanding. These characteristics may link to a sense of belonging and creation of common ground between the participants.

## Adaptation

Few adaptations were considered to be needed to the role of the expert mother, following the feasibility study, excepting the increased mentoring on providing mental health support,

already described. Weekly mentoring guidelines were developed by the research team, and in partnership with the expert mothers and facilitator therapists, after the first pilot. These guidelines were used to assist the research assistants (psychologists) to provide targeted mentoring for the facilitators to support emotional wellbeing of participants. Resources and processes were refined through the piloting of the 'Juntos' intervention and sections of the paper based manual were transformed to video format. The videos were reported as helpful to expert mothers to further demonstrate techniques to participants. Additional resources were requested by the participants and these included:

> *'We could bring pictures of our shower chairs, or maybe you could show us pictures with the different phases of communication and development of eating, so that we can identify where our children are now, and what the next steps will be–just like we did for the development of moving."* (Participant 08)

There was a need for the study co-ordinator to provide support to facilitators in organisation of logistics e.g. travel, food and intervention materials. The study co-ordinator also provided instruction, in addition to the comprehensive intervention manual, on how to run practical sessions in order to maximise their impact. Therefore, more practical support may be needed in organizing the sessions in future programmes.

When considering future scale up in Brazil, it is likely that adaptation of the training to include a greater number of expert mothers compared to the number of facilitator therapists may be required to account for contexts with fewer number of therapists and as mothers may not have the capacity to travel as much or facilitate as many groups as the therapists. The 'Juntos' intervention performed in a similar way with all three groups, which consisted of a different population in each group. The co-ordinator selected different sites for the groups and it was reported as important for the expert mother to be local to the community in which the community-based group intervention is run, whilst it is acceptable for the facilitator therapist to travel further and be involved in a greater number of groups.

## Discussion

The theory of change guiding the 'Juntos' intervention is that a sense of belonging and creation of common ground would provide an environment to improve the knowledge and skills of caregivers through a social support network. This approach, in turn, would improve the quality of life of children with developmental disabilities and their caregivers, as indeed has shown to be the case for a similar intervention [26]. We introduced the expert mothers as facilitators in order to reinforce the participatory and peer learning aspects. Our findings show that the role of mothers as facilitators in community group interventions is likely to be a feasible approach to participatory peer learning to improve care for children with developmental disabilities. The use of expert mothers was considered to be acceptable for participants and facilitator therapists and there was demand for their role.

### Comparison to other studies

Our findings contribute to the growing body of evidence of the importance of participatory peer learning to improve child care and support caregivers' psychological and emotional wellbeing [29, 26]. A systematic review and meta-analysis of randomised controlled trials undertaken in Bangladesh, India, Malawi, and Nepal of women's groups practicing participatory learning and action show that these practices improve maternal and neonatal survival [30], by increasing appropriate care-seeking, home prevention and care practices for mothers and

newborns [31]. The women's groups drew on principles of Paulo Freire's work [32], namely: (i) health challenges are often rooted in powerlessness, and can be addressed by social empowerment; (ii) including dialogue and problem solving in health education is more empowering than information giving; and (iii) communities can develop critical consciousness to recognise and address the underlying social and political determinants of health. While there is no single recognised theory of how women's groups practising participatory learning and action achieve their health impacts [33], group participation and membership offers a valuable social support network to navigate medical hierarchies, and may contribute to change in care practices through increasing confidence of caregivers.

Comparison of our findings is also possible with other interventions for children with developmental disabilities in different low resourced settings, such as: (i) a Caregiver Skills Training (CST) for caregivers of children with intellectual disabilities [34, 35], developed by the World Health Organisation, (ii) 'Titukulane', an eight module community group intervention that aims provide contextualised psychological support to caregivers of children with intellectual disabilities [36], (iii) Learning through Everyday Activities with Parents (LEAP-CP), which aims to improve the mobility of children with cerebral palsy over 30 weekly peer-to-peer home visits [37], and (iv) PASS, a parent-mediated intervention for autism spectrum disorder in India and Pakistan [38] that was adapted for delivery by non-specialist workers and uses video feedback methods to address parent–child interaction. The focus on caregiver involvement is a common thread in all of these interventions, which is critical, particularly where there are few health services. These interventions demonstrate that reaching family-centred care goals can be facilitated through having mothers as facilitators. However, formal evaluation of their effectiveness and cost-effectiveness is lacking.

## Value added from expert mothers

A scoping visit by the research team to Salvador, Bahia, prior to piloting the intervention, demonstrated that the majority of support that was provided to families was medically orientated and that informal support networks that were established varied in focus and structure [39]. For example, some groups focussed on advocacy and promoting children's rights, while many mothers reported being part of WhatsApp groups with other caregivers, which provided some social and emotional support on an ad hoc basis. The addition of an expert mother to the 'Juntos' intervention therefore supports the provision of family-centred services through including the sharing of lived experience of caring for a child with CZS. In this study, the role of expert mothers was seen specifically as being important to share and learn together, and to provide support and encouragement.

## Limitations

This study has limitations. We explored feasibility of use of two expert mothers in only one setting. We did not compare different strategies of delivery. Nor did we explore in detail what aspects of expert mother were critical to success. Consequently, it is difficult to identify the extent to which the perceived feasibility depended on specific personal characteristics of individual expert mothers, and how much on the use of an expert mother of any kind. The therapist and expert mothers were paid and equal amount and this may have influenced their strong commitment to the programme. Only a subset of participants was selected to participate in semi-structured interviews and this selection, although purposive to gather a range of perspectives, may have introduced a positive reporting bias in the responders. Attendance rates may have influenced participant responses although there was no evidence of better retention for earlier compared to later modules. Furthermore, the selection of interview

participants, although done to gather a range of perspectives, may have brought an inherent bias to the qualitative data that was analysed from the participants. However, the triangulation of findings between participants, expert mothers and facilitator therapists gives us confidence in our results. In addition, with a small sample size it is not possible to draw any firm conclusions with regards limited efficacy of the programme or the impact on families of children with different severities of functional impairment.

## Implications and future steps

We have identified research questions that have been framed by the gaps in evidence of this feasibility study. Future studies may seek to establish the cost-effectiveness and long- term benefits (such as improved survival and hospitalization rates) of the inclusion of expert mothers in the delivery of the Juntos intervention. Future research should also investigate the impact of the personal characteristics and experience of the person offering support, and the impact of peer support on caregivers' relationships with health care professionals. In addition, the role of equal payment between the two facilitators warrants further attention.

## Conclusion

Caregivers with similar life experiences may provide innovate community support to families of children with CZS in resource limited settings. The use of expert mothers in a participatory group setting offers a unique approach to harness the capacity of families to provide care for their child and may be feasible in similar settings. Future consideration for scale up in Brazil includes accounting for resource-limited contexts with fewer number of therapists. It is likely that adaptation of the training to include a greater number of expert mothers compared to the number of facilitator therapists may be required.

## Supporting information

**S1 Table. Post intervention interview guide–facilitators.**
(DOCX)

**S2 Table. Post intervention interview guide–participants.**
(DOCX)

## Author Contributions

**Conceptualization:** Tracey Smythe, Julia Reis, Antony Duttine, Hannah Kuper.

**Data curation:** Tracey Smythe.

**Formal analysis:** Tracey Smythe, Monica Matos.

**Funding acquisition:** Hannah Kuper.

**Investigation:** Tracey Smythe, Monica Matos, Julia Reis, Antony Duttine.

**Methodology:** Monica Matos, Antony Duttine, Silvia Ferrite.

**Supervision:** Silvia Ferrite, Hannah Kuper.

**Writing – original draft:** Tracey Smythe.

**Writing – review & editing:** Tracey Smythe, Monica Matos, Julia Reis, Antony Duttine, Silvia Ferrite, Hannah Kuper.

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
