## [Decision Letter · Decision Letter 0]

5 May 2020

PONE-D-20-02401

Mothers as facilitators for a parent group intervention for children with congenital zika syndrome: qualitative findings from a feasibility study in Brazil

PLOS ONE

Dear Dr Smythe,

Thank you for submitting your manuscript to PLOS ONE. After careful consideration, we feel that it has merit but does not fully meet PLOS ONE’s publication criteria as it currently stands. Therefore, we invite you to submit a revised version of the manuscript that addresses the points raised during the review process.

ACADEMIC EDITOR: You will see that both reviewers have provided very detailed feedback and suggestions on how to improve each section of the manuscript. They are agreed that the article could be significantly improved and should then be resubmitted, as the topic is relevant and may contribute to improving the case of children who suffer from CZS in Brazil and other low- and middle-income countries. Good luck with making these revisions.

We would appreciate receiving your revised manuscript by Jun 19 2020 11:59PM. To enhance the reproducibility of your results, we recommend that if applicable you deposit your laboratory protocols in protocols.io, where a protocol can be assigned its own identifier (DOI) such that it can be cited independently in the future. For instructions see: http://journals.plos.org/plosone/s/submission-guidelines#loc-laboratory-protocols

We look forward to receiving your revised manuscript.

Kind regards,

Shelina Visram, PhD, MPH, BA

Academic Editor

PLOS ONE

Journal Requirements:

2. Please address the following:

- Please ensure you have thoroughly discussed all potential limitations of this study within the Discussion section, including potential biases introduced by using self-reported data.

- Please include additional information regarding the interview guide used in the study and ensure that you have provided sufficient details that others could replicate the analyses. For instance, if you developed a guide as part of this study and it is not under a copyright more restrictive than CC-BY, please include a copy, in both the original language and English, as Supporting Information.

'I have read the journal's policy and the authors of this manuscript have the following

competing interests:One of the authors (AD) joined the Pan American Health

Organisation (PAHO) during the research period. Work on the research study was

undertaken outside and separate to his PAHO duties.'

Additional Editor Comments (if provided):

Reviewers' comments:

Reviewer's Responses to Questions

**Comments to the Author**

1. Is the manuscript technically sound, and do the data support the conclusions?

Reviewer #1: Partly

Reviewer #2: Partly

2. Has the statistical analysis been performed appropriately and rigorously? 

Reviewer #1: N/A

Reviewer #2: N/A

3. Have the authors made all data underlying the findings in their manuscript fully available?

Reviewer #1: Yes

Reviewer #2: No

4. Is the manuscript presented in an intelligible fashion and written in standard English?

Reviewer #1: No

Reviewer #2: Yes

5. Review Comments to the Author

Reviewer #1: The topic of the article is relevant and contributes to the literature on Zika. However, the authors need to review detailed edits and comments suggested in the file attached. The methods, results, discussion and conclusion sections need to change significantly because there is confusion on what should be part of each section.

There are several definitions of categories that are included in the results section that should be moved to methods. The results section should incorporate a more in-depth analysis of the findings to substantiate the evidence provided in the practicality, adaption and implementation categories. The acceptability category is much better supported by the evidence provided than the other four categories. I assume that there are more data (narratives from lay experts and participants) to support the analyses of those.

Also, the discussion section should not include results but instead discuss them according to the literature on the subject, which is partially done. There may be other relevant studies that could enrich the discussion of the results.

Once changes are incorporated in this article, the abstract has to reflect those appropriately. The sentence "One expert mother was unable to attend two sessions due to her child being ill" is a small detail that should not be part of the abstract, for example.

In addition, you need to be more specific in your conclusions, which are too vague. You need to clarify and be more specific about how the use of lay experts could be implemented in Brazil and other low and middle-income countries to address the developmental disabilities of CZS.

Reviewer #2: In this study, the authors use qualitative research methodology to assess the feasibility of a community based intervention that includes using “expert mothers”, whose children also have congenital zika syndrome (CZS), as part of a team to educate families of children with CZS through a series of educational modules. The authors report that the intervention was feasible and that caregivers and expert mothers had favorable opinions on this intervention. This line of research is important because children with CZS may have severe cognitive, speech, and motor delays and developing strategies to educate caregivers is essential.

This article benefits from the qualitative research approach and the in-depth responses that were gained from participants. However, the authors could further strengthen this manuscript by a more in-depth discussion of the patient population, the generalizability of this approach in other settings where congenital zika virus exposure occurred, and limitations of the current intervention. My specific revision suggestions are outlined below:

1. The number of parents/caregivers utilized in the study and the actual number of completed questionnaires/interview is somewhat ambiguous. The results section states that 27 families were enrolled, which yielded 38 participants who complete at least one session. They then state that 30 focus groups, 9 participant interviews, and 4 facilitator interviews were conducted. This section would benefit from a more in-depth clarification of the patient population and retention rates:

a) Although 38 participants completed at least one session, how many participants (and families) actually completed the full 10 module sessions? What was the average number of sessions attended by families?

b) What were the attendance rates for different modules and were there better retention for earlier vs later modules. i.e. was there any bias in terms of what information participants were exposed to and could this have affected their responses?

c) How many of the 38 participants completed focus groups and interviews? Results state that there were 30 focus groups, 9 participant interviews, and 4 facilitator interviews. This does not match the numbers listed in “Study Design” section on page 8 were they report 26 participants in focus groups and 13 interviews. Clearly stating the number of responders (and families) would be helpful in understanding the scope of the data.

d) Were participant interviews done individually, so were only 9 out of the 38 participants actually interviewed?

2. The severity of a child’s speech, fine motor, and gross motor disabilities will affect the usefulness of specific training modules. For example, a module on “managing seizures” may not be useful for a family who’s child does not have seizures. Similarly a module on “feeding challenges” will be very different for a child who is fed through a gastrostomy tube vs a child who is fed orally. Additionally the educational and socio-economic status of families will have a tremendous impact on your results. As such, it would be helpful to better understand your patient population in terms of basic patient characteristics:

a) What proportion of children had mild vs moderate vs severe delay?

b) What was the prevalence of epilepsy, feeding difficulties, microcephaly, etc. in this patient cohort?

c) The study states that participants were literate but what was the socioeconomic status of participants and their home/neighborhood environment? Was it mostly patients living in poverty or more middle-class families?

I fully understand if some of this information is not available but having some sense of the patient population would help others understand what groups of patients would most benefit from the proposed approach.

3. Further information on the “expert mother” role would be helpful. Specifically (although the results and discussion sections touch on these topics), the methods section should specify how they were identified, what sort of training they received prior to the teaching sessions, and how often they were from the same neighborhood as the participants.

4. The results section does not describe any concerns or criticisms families or expert mothers may have had about the intervention. The article would thus benefit a breakdown of any negative reactions/limitations categorized by the five elements of feasibility that the authors utilize in their analysis. If there were no negative reactions, there should be a discussion of why that is the case and if there is a positive bias among the responders.

5. The discussion section would benefit from a succinct and explicit discussion on the “value added” from expert mothers as it pertains to children with CZS. Are there prior zika-related community interventions that were done without expert mothers? If so, what benefits do you think the addition of an expert mother brings.

6. The discussion section would benefit from a summary of the challenges encountered in implementing the intervention (for example expert mothers not being able to attend, as they describe earlier in manuscript) and approaches the authors took to address these challenges. This would be beneficial for others attempting to implement such interventions.

7. Although the discussion cites other studies where similar approaches have improved patient outcomes, whether the current intervention had long term benefits and improves survival, hospitalization rates, and other outcomes is unclear. Additionally, short term interventions can often suffer from lack of long-term benefits due to their limited nature. The discussion would benefit from expanding the “limitations” paragraph to address these issues and potentially offer suggestions on how the authors could alleviate these limitations moving forward.

6. PLOS authors have the option to publish the peer review history of their article (what does this mean?). If published, this will include your full peer review and any attached files.

Reviewer #1: No

Reviewer #2: No

---

## [Author Response · Author response to Decision Letter 0]

5 Jun 2020

Please see attached 19 page letter of response.

---

## [Decision Letter · Decision Letter 1]

7 Aug 2020

PONE-D-20-02401R1

Mothers as facilitators for a parent group intervention for children with Congenital Zika Syndrome: qualitative findings from a feasibility study in Brazil

PLOS ONE

Dear Dr. Smythe,

Thank you for submitting your manuscript to PLOS ONE. After careful consideration, we feel that it has merit but does not fully meet PLOS ONE’s publication criteria as it currently stands. Therefore, we invite you to submit a revised version of the manuscript that addresses the points raised during the review process.

ACADEMIC EDITOR: You will see below that both reviewers feel that all of their comments have now been addressed, so well done. However, we are sending this paper back to you for further revision because R1 and I feel that the spelling and grammar needs some work. This should not take long to address - please have the article proofread and get it back to us as soon as you can. Once it comes back to me I can take a quick check before making the decision to accept without sending it back out to the peer reviewers.

We look forward to receiving your revised manuscript.

Kind regards,

Shelina Visram, PhD, MPH, BA

Academic Editor

PLOS ONE

Reviewers' comments:

Reviewer's Responses to Questions

**Comments to the Author**

1. If the authors have adequately addressed your comments raised in a previous round of review and you feel that this manuscript is now acceptable for publication, you may indicate that here to bypass the “Comments to the Author” section, enter your conflict of interest statement in the “Confidential to Editor” section, and submit your "Accept" recommendation.

Reviewer #1: All comments have been addressed

Reviewer #2: All comments have been addressed

2. Is the manuscript technically sound, and do the data support the conclusions?

Reviewer #1: Yes

Reviewer #2: Yes

3. Has the statistical analysis been performed appropriately and rigorously? 

Reviewer #1: N/A

Reviewer #2: N/A

4. Have the authors made all data underlying the findings in their manuscript fully available?

Reviewer #1: Yes

Reviewer #2: Yes

5. Is the manuscript presented in an intelligible fashion and written in standard English?

Reviewer #1: Yes

Reviewer #2: Yes

6. Review Comments to the Author

Reviewer #1: Small edits are needed to standardize writing of some words, such as newborns, ongoing, etc. Adaption is not a word in English. It should be Adaptation. The article still needs revision of the English to fix a few issues.

Reviewer #2: The authors have addressed my concerns, I have not further revision recommendations or other concerns with the manuscript in current form.

7. PLOS authors have the option to publish the peer review history of their article (what does this mean?). If published, this will include your full peer review and any attached files.

Reviewer #1: **Yes: **Carlos Eduardo Siqueira

Reviewer #2: No

---

## [Author Response · Author response to Decision Letter 1]

10 Aug 2020

Please see full response to reviewers in attached documentation.

---

## [Editor Report · Decision Letter 2]

26 Aug 2020

Mothers as facilitators for a parent group intervention for children with Congenital Zika Syndrome: qualitative findings from a feasibility study in Brazil

PONE-D-20-02401R2

Dear Dr. Smythe,

We’re pleased to inform you that your manuscript has been judged scientifically suitable for publication and will be formally accepted for publication once it meets all outstanding technical requirements.

Kind regards,

Shelina Visram, PhD, MPH, BA

Academic Editor

PLOS ONE
---

## [Editor Report · Acceptance letter]

28 Aug 2020

PONE-D-20-02401R2 

Mothers as facilitators for a parent group intervention for children with Congenital Zika Syndrome: qualitative findings from a feasibility study in Brazil 

Dear Dr. Smythe:

I'm pleased to inform you that your manuscript has been deemed suitable for publication in PLOS ONE. Congratulations! Your manuscript is now with our production department. 

Kind regards, 

on behalf of

Dr. Shelina Visram 

Academic Editor

PLOS ONE